# Recent Advances of Microfluidic Platform for Cell Based Non-Invasive Prenatal Diagnosis

**DOI:** 10.3390/ijms24020991

**Published:** 2023-01-04

**Authors:** Hei-Jen Jou, Pei-Hsuan Lo, Pei-Ying Ling

**Affiliations:** 1Departments of Obstetrics and Gynecology, Taiwan Adventist Hospital, Taipei 105404, Taiwan; 2School of Nursing, National Taipei University of Nursing and Health Science, Taipei 112303, Taiwan; 3Departments of Obstetrics and Gynecology, National Taiwan University Hospital, Taipei 100225, Taiwan; 4International College of Semiconductor Technology, National Yang Ming Chiao Tung University, Hsinchu 300093, Taiwan

**Keywords:** non-invasive prenatal diagnosis, circulating fetal cells, extravillous trophoblasts, fetal nucleated RBCs, microfluidic devices, immunoaffinity, size-based microfluidics

## Abstract

The purpose of the present review is to try to highlight recent advances in the application of microfluidic technology on non-invasive prenatal diagnosis (NIPD). The immunoaffinity based microfluidic technology is the most common approach for NIPD, followed by size-based microfluidic methods. Immunoaffinity microfluidic methods can enrich and isolate circulating fetal extravillous trophoblasts (fEVTs) or fetal nucleated red blood cells (fnRBCs) for NIPD by using specific antibodies, but size-based microfluidic systems are only applied to isolate fEVTs. Most studies based on the immunoaffinity microfluidic system gave good results. Enough fetal cells were obtained for chromosomal and/or genetic analysis in all blood samples. However, the results from studies using size-based microfluidic systems for NIPD are less than ideal. In conclusion, recent advances in microfluidic devices make the immunoaffinity based microfluidic system potentially a powerful tool for cell-based NIPD. However, more clinical validation is needed.

## 1. Introduction

Currently, 1st-trimester chorionic villus sampling (CVS) and 2nd-trimester amniocentesis are still the tests of gold standard for diagnosing fetal chromosomal abnormalities. However, each test carries a risk of fetal loss, which is about 1 in 1000 for amniocentesis and about 1 in 500 for CVS. Therefore, they were suggested to be only performed under certain indications, including advanced maternal age, a family history of chromosomal aberrations or a positive result on non-invasive screening tests. Although the fetal loss rates are not high, it is unavoidable. The unwanted risk of fetal loss prompts the development of a non-invasive prenatal test. Current non-invasive prenatal test (NIPT) based on cell free fetal DNA (cffDNA) still requires further invasive testing to confirm the diagnosis in pregnant women with a positive test result. The purpose of this article is to review recent studies on the application of microfluidic devices for the isolation of circulating fetal cells (CFCs), and to explore their clinical feasibility for non-invasive prenatal diagnosis (NIPD). Only studies that utilize microfluidic devices to enrich and isolate CFCs, and followed by chromosomal and/or genetic analysis for NIPD, are included in this review.

## 2. The Progress of Prenatal Screening Test for Fetal Aneuploidies

Over the past decades, advanced maternal age was used as an indication for invasive procedures (amniocentesis and chorionic villus sampling) for the diagnosis of fetal aneuploidies. However, the sensitivity is quite low with a high false positive rate (FPR) by using maternal age as a screening index. Furthermore, more than 70% of newborns with Down syndrome are born to mothers under the age of 35 [1]. Therefore, many efforts had been made to try to find more reliable screening tools.

Screening tests using maternal serum biomarkers were then developed. With a positive screening rate of 5%, the detection rates of fetal Down syndrome were about 69% for triple test and 81% for quadruple test, respectively [2]. In the 1990s, an ultrasound marker, nuchal translucency measurement, was introduced as a screening tool for fetal aneuploidies [3]. With a combination of the measurement of nuchal translucency and serum tests, the detection rates for fetal triploidy at a 4% FPR can reach as high as 90%, 97%, and 92% for trisomy 21, trisomy 18, and trisomy 13, respectively [4].

In 1997, Lo et al., reported the detection of fetal DNA in maternal peripheral blood [5]. This discovery ushered in a new era of NIPT; cffDNA-based NIPT has been in commercially available and widely used since 2011 [5]. The cffDNA-based NIPT is a non-invasive procedure with high sensitivity and specificity for the detection of fetal aneuploidies, including trisomy 21, trisomy 18, trisomy 13, and monosomy X [6,7].

However, NIPT is a screening test rather than a diagnostic test. High-risk mothers by NIPT require a further invasive procedure to obtain a definitive diagnosis. The accuracy of NIPT is also affected by many factors, such as fetoplacental mosaicism, non-identical vanishing twins, or maternal malignancy [8].

Therefore, many studies have attempted to isolate fetal cells from the maternal blood of pregnant women for chromosomal and genetic analysis. Because fetal cells have complete chromosomes and genes, NIPD can be achieved without additional invasive tests to confirm the diagnosis of fetal chromosomal or genetic anomalies. Nevertheless, it remains a big challenge for researchers to isolate enough fetal cells from maternal blood for chromosomal or genetic analysis, due to technical difficulties. This dilemma has not brought a glimmer of light until the development of microfluidic chips, and the advancement of genetic analysis technology, in recent years.

## 3. Cell Based NIPD

Walknowska et al., first reported the discovery of fetal lymphocytes with male karyotype in maternal blood [9]. Since then, how to isolate fetal cells from maternal blood for analysis as a non-invasive prenatal diagnosis has become the “Holy Grail” of prenatal testing. There are several types of fetal cells that can be isolated from the peripheral blood of pregnant women [10]. Among them, the most studied cell types are fetal extravillous trophoblasts (fEVTs), or fetal nucleated red blood cells (fnRBCs).

fEVTs are cells which originate and differentiate from anchoring villi that attach to the uterine wall, and can be detected in the peripheral blood of pregnant women during the first trimester [10]. Therefore, many studies have used fEVTs as target cells for NIPD. The advantages of utilizing fEVTs for NIPD include: (1) the life cycle of fEVTs is rather short, perhaps only a few days. Therefore, it is unlikely to detect fEVTs from pre-pregnancy; and (2) fEVTs have many specific biomarkers (eg: cytokeratins, CD105, H315, HLA-G, GB17, GB25, CD141, etc.) that can be used to enrich and identify fEVTs [11]. On the other hand, the use of fEVTs also has disadvantages as the target of NIPD, including: (1) the number of fEVTs in maternal blood is quite rare, especially in the first trimester. Therefore, it is necessary to have high-efficiency capture technology and separation technology, or need more blood volume in order to obtain enough fEVTs for NIPD detection; and (2) the source of fEVTs is the placenta. Chromosomal mosaicism was detected in 2.18% of CVS. The overall risk of true fetal mosaicism is 13% [12].

fnRBCs are other target cells for NIPD. Kleihauer et al., reported the detection of fetal hemoglobin in the red blood cells of a blood smear from the peripheral blood of a pregnant woman in 1957 [13]. fnRBCs can be detected in the blood of pregnant women as early as the fifth week of pregnancy. The number of fnRBCs reaches as high as 50,000 per milliliter of blood in the 12th week of pregnancy, and then drops to 1000 per milliliter by the 20th week of pregnancy [14]. The lifespan of fnRBCs is between 25 to 35 days [14,15,16]. The short lifespan makes fnRBCs less likely to survive until the next pregnancy. The estimated number of fnRBCs ranges from 1 in 10^5^ to 1 in 10^9^ cells in maternal blood [14,17,18]. Table 1 summarized recent studies of cell based NIPD by using microfluidic devices.

Since the first detection of fetal cells from maternal blood by Walknowsk et al., many studies have attempted to isolate CFCs for NIPD using various non-microfluidic approaches. However, the results of these studies are not sufficient to support the clinical application of cell based NIPD, because of the low recovery rate of CFCs [10,27]. With the advances in enrichment technology of circulating rare cells (CRCs) over the past few years, research related to CRCs have increased rapidly, even reaching about 2000 articles per year on PubMed. Most of them are related to circulating tumor cells (CTCs), and there are only very few studies focus on CFCs. There are several reasons why CFCs are less studied and the technique related CFCs progressed late. Compared with the technologies for CTCs, NIPD needs more stringent requirements for cell enrichment, identification, and isolation, which can provide the tests of higher sensitivity, specificity, and efficiency, as well as extremely low false positives. Therefore, the research on cell based NIPD began to increase only after several years, since the development of microfluidic technology.

We used “microfluidic” and “non-invasive prenatal diagnosis”, “circulating fetal cells”, “circulating trophoblast”, or “fetal nucleated RBC”, as keywords to search for literature on PubMed and obtained the full text of the literature. After careful reading, articles with laboratory tests but in absence of NIPD verification were excluded. Table 1 summarized recent studies of cell based NIPD by using microfluidic devices.

## 4. Microfluidics Technologies for NIPD

Various techniques have been developed to enrich and identify circulating rare cells (CRCs), including CTCs and CFCs. The decade from 1993 to 2003 was the first “golden era” of circulating fetal cells [28]. Many studies have attempted to isolate fetal cells from maternal blood for NIPD, with techniques involving MACs, FACs, or density gradient centrifugation, etc. [10]. However, this trend of cell-based NIPD has not developed a clinically applicable technology because of the inability to effectively separate fetal cells from maternal blood and the high false positive rate [29].

In the past decade, microfluidic technology has developed more rapidly because of its many advantages, such as high throughput, low cost, high efficiency, precision and low sample consumption [30]. For example, Wang et al., have reported a capture efficiency about 81% and a high purity of 83% by using multifunctional microsphere-assisted inertial microfluidics to isolate fnRBCs [31]. The main principle used for current CRC enrichment techniques to separate target cells from back-ground cells is based on the physical, biological or both properties of the target cells. Physical techniques exploit differences in physical properties of target cells and back-ground cells (including cell size, density, charge, and deformability) to separate the two. For a biological approach, antibodies against specific markers of fetal cells were used to isolate fetal cells from background cells [32]. The immunoaffinity technique is the most common method of biological approach [33].

### 4.1. Immunoaffinity Based Microfluidic Technique

Recent advances in nanomaterials and fabrication technology of nanostructured biochips provide considerable advantages to detect and isolate CRCs by using immunoaffinity based nanostructured microfluidic devices with high purity, high sensitivity, and high efficiency [33]. The immunoaffinity approach is the most common microfluidic method to isolate CFCs for NIPD. Nearly half of these studies targeted fnRBCs for NIPD [19,22,23], and the other half targeted EVTs [20,24,25,26]. Only one study performed analysis of both fnRBCs and fEVT [21]. In 2017, three different research groups reported their achievement in cell based NIPD.

In May 2017, He et al., in Wuhan University reported the use of microfluidic chips to isolate fnRBCs from maternal blood for NIPD. They used biocompatible nanoparticles to form 3D nanostructure on the chip. The chip was then coated with a thin layer of anti-CD147 antibodies to catch fnRBCs when the blood sample flow through the chip. The fnRBCs were identified by immunocytochemical (ICC) staining with ε-HbF+/CD71+/DAPI+. On-chip FISH was then performed to successfully diagnose seven fetuses with chromosomal trisomies [19]. In 2018, another nanostructured biochip, biotin-doped Ppy microchip, was used to enrich cfNRBC for NIPD by same study group with similar enrichment and identification workflow. An electrical stimulation method was use to release fnRBCs form the microchip. The diagnosis of fetal chromosomal aneuploidies in 12 pregnancies was made by subsequent FISH on released fnRBCs, and one case of microdeletion syndrome was diagnosed by WEA of the enriched fnRBCs [22].

Hou et al., published their results of cell based NIPD in 2017 by using imprinted NanoVelcro microfluidic chips. The surface of the chip is coated with a layer of anti-EpCAM antibodies. When the blood sample passes through the microfluidic channel of the chip, fEVTs are captured by the chip because of the interaction of the anto-EpCAM antibody on the surface of the chip and antigens on the cell surface. The fEVTs were identified by ICC staining with Hoechst+/CK7+/HLA-G+/CD45− and a cell size between 12 and 20 μm. Laser capture microdissection (LCM) was used to perform single-cell biopsy to isolate single fEVTs from the chip. In this study, 3–6 fEVTs and 5–15 fEVTs were obtained from per 2 mL of peripheral blood of healthy and diseased cohorts, respectively. Subsequent genetic analysis on the isolated fEVTs successfully detected chromosomal aberrations on nine fetuses [20].

In the same year, Huang et al., reported their results for NIPD using a novel microfluidic chip in 25 pregnant women (including five who underwent CVS or amniocentesis for genetic and chromosomal analysis). PicoBioChip is a nanostructured chip coated with a thin layer of biotin-streptavidin complex. Blood samples are pre-treated with biotinylated antibodies, allowing the biotinylated antibodies adhere onto the cell surface. When the blood sample flows through the PicoBioChip, the target cells were then captured by the chip through the strong binding power force of biotin-streptavidin interaction. In this study, an attempt to isolate both fnRBCs and fEVTs for NIPD was performed. Biotinylated anti-CD71 antibodies and biotinylated anti-EpCAM antibodies were used as enrichment antibodies for fnRBCs and fEVTs, respectively. Count-in/filter-outer criteria according to ICC staining was used to identify fnRBCs and fEVTs with CD71+/GPA+/CD45−/DAPI+ for fnRBCs and CK7+/HLA-G+/CD45−/DAPI+ for fEVTs, respectively. The captured cell counts were 1–44 fnRBCs/2 mL and 1–32 fEVTs/2 mL in the validation group. Enough fnRBCs were obtained for successful FISH and WGA analysis in all the five validation cases. However, only four FISH and two WGA analyses were successful in the five cases when EVTs were used for NIPD because that not enough fEVTs were caught [21]. A report published two years later by the same research group described the application of a silicon-based coral-like nanostructured microfluidics, which was optimized from PicoBioChip, to enrich fRBCs for NIPD in 14 pregnant women. In addition to Coral Chips, they also introduced a new automated Cell Picker in place of manual operations to accurately isolate fnRBCs in their research. Sufficient fnRBCs (2–71 fnRBCs/2 mL of blood) were successfully isolated from all blood samples to achieve chromosomal and genetic analysis and genetic analysis [23].

Sonek et al., used an automated microfluidic-based LiquidScan^®^ machine to enrich cfEVTs and compared the efficiencies of two different enrichment antibodies, anti-EpCAM antibodies and a proprietary antibody mixture. More cfEVTs were obtained by antibody mixture than anti-EpCAM antibodies with an average of 2.67 cells vs. 2.11 cells from per mL of blood, respectively. Nine cases of fetal aneuploidies were successfully diagnosed by FISH on the isolated cfEVTs [24].

Figure 1 summarizes the experimental procedures of previous studies for cell based NIPD using the microfluidic technology. There are two different principles of immunoaffinity approach were used for the enrichment of fnRBCs or fEVTs. One relies on the interaction of biotin-streptavidin biological bonds [21,23,34,35], and the other is based on the interaction between antibody and antigen [19,20,22,24]. The design and fabrication of nanostructures of the chips aims to increase the contact area between chip and cells in order to increase the efficiency of cell capturing as well as maintain the integrity of cell morphology.

### 4.2. Size-Based Microfluidic Technology

Size-based isolation is the oldest approach of CRC enrichment, the principle of which is to separate the target cells from background cells according to the size and deformability of the cells when the blood sample flows through a membrane-like filtration system. Advances in the fabrication of microfluidic systems can greatly increase the efficiency of CRC enrichment by controlling pore sizes and nanostructure on the microfluidic chip through precise fabrication techniques. There are several size-based microfluidic devices used to isolate fEVTs, including Dielectrophoresis, (DLD), inertial and lamina vortex technologies [36].

Caryrefourcq et al., used the DEPArray method to isolate fEVTs and used Huntington disease as a clinical model for cell-based NIPD in 2020. Among the seven participants, only five fEVTs were isolated from four pregnant women and only one got a conclusive NIPD [25].

In 2020, Huang Y et al., reported the use of a size-based four-stage inertial microfluidic device to separate fEVTs for NIPD [26]. The authors emphasize that this is a fast and low-cost laboratory process to enrich fEVTs. The fEVTs can be isolated from 2cc of whole blood of pregnant women within 5 min without pretreatment, with an average enrichment efficiency of 52.3–65.8% and removal of 99.95% of white blood cells. After enrichment of target cells, fEVTs were identified by immunofluorescent staining of CK7+/CD45−/DAPI+. The study revealed that fEVTs could be isolated in 26 of 30 pregnant women and followed by downstream single genetic analysis. Although the process of enriching fEVTs is timesaving, the experimental steps after enrichment are labor-intensive and may cause cell loses [26].

## 5. Isolation of Fetal Cells

Due to advances in fabrication technique of microfluidic system, recovery rates of CRCs can reach 80–90% or higher for most systems [30]. Chromosomal aberrations can be detected using on-chip FISH. However, the fetal cells must be isolated or release from the chips before further downstream genetic analysis. It remains a big challenge to effectively and precisely isolate fetal cells from the microfluidic chips. A large number of target cells may be lost during the isolation or detachment process. Therefore, an effective isolation approach is needed in addition to an efficient enrichment procedure in order to get enough fetal cells for downstream genetic analysis. Because of the limitations of cell isolation techniques, most studies can only perform genetic analysis on a cluster of cells, rather than a single cell or a bulk of specific cells [37]. The result of genetic analysis of fetal cells may be affected by the large amount of contaminated maternal cells in the enriched cell samples.

For the first time, Feng et al., introduced the use of electrical stimulation to release fnRBCs from the microfluidic chips. Higher current stimulation with 0.8 V for 15 s can release up to 94.6% of fnRBCs [22].

The combination of the NanoVelcro system and LCM has been shown to have the ability to isolate high-purity CTCs from the blood of prostate cancer patients [38] Hou’s study showed that cfEVTs can also be successfully isolated in the same way [21]. LCM is a powerful technique to isolate target cells from background cells. The mechanism of LCM is to use a transparent thermoplastic membrane to adhere to the target cells, and then use low-energy infrared laser pulse to melt the structure around the target cells to isolate the target cells [39]. The advantage of the LCM approach is the ability to isolate a single fetal cell with high purity. However, the process of LCM needs to be operated under direct microscope visualization, which is quite labor-consuming.

The manual operation of cell isolation process is time-consuming and labor-intensive, which may severely limit the throughput of the tests. Therefore, it is necessary to develop an automated cell isolation system. An automatic Cell Picker machine was introduced in Ma’s study, which was integrated with an automatic image recognition and positional system [23]. Under the control of software, the system can automatically identify the target cells and then perform single cell biopsy. In addition to being able to precisely isolate a single target cell, it also has the advantage of timesaving and laborsaving.

## 6. Conclusions

In this review, we described the recent advances in cell-based NIPD. While invasive tests, including amniocentesis and chorionic villus sampling, remain the most reliable modality for diagnosing fetal chromosomal and genetic abnormalities, their possible side effects have prompted the development of NIPD. With the advancement of microfluidic chip design and manufacturing, there are many microfluidic chip systems that have demonstrated their effectiveness in capturing CFCs. Combined with single-cell biopsy technology, CFCs can be effectively isolated for downstream chromosome and genetic testing. According to the limited studies, the immunoaffinity technique has the greatest potential to be developed as a NIPD tool. Label-free methods that rely on the physical properties of cells cannot not provide sufficient evidence to effectively isolate CFCs for NIPD. Although the preliminary findings are quite promising, a large number of studies are still needed to confirm the feasibility of its clinical application at this stage. In the next few years, there will be more research on NIPD, which also means that we are one step closer to the “Holy Grail” of NIPD.

## Figures and Tables

**Figure 1 ijms-24-00991-f001:**
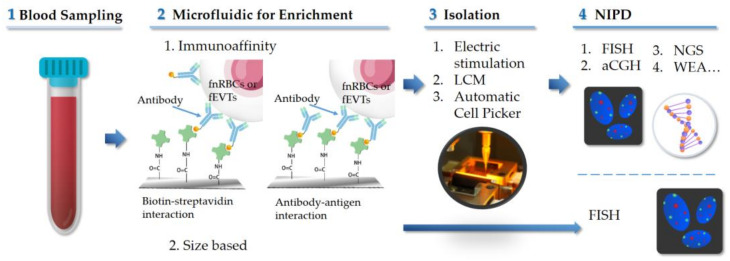
Schematic of laboratory process of NIPD based on microfluidic technology. 
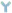
: antibody; 

: biotin; 
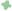
: streptavidin; LCM: laser capture microdissection.

**Table 1 ijms-24-00991-t001:** A summary of microfluidic technical approaches for detection and isolation of fetal cells from maternal peripheral blood.

Authors/Year/Ref.	Case No.	GA	Cell Type	Microfluidic Platform	Markers for Enrichment	Immuno-Staining	Cell No.	Down-Stream Analysis
Immunoaffinity								
He (2017) [19]	48	10–30	fnRBCs	HA/CTS NPs	CD147	ε-HbF+/CD71+/DAPI+	3–71/mL	FISH
Hou (2017) [20]	15	8–23	EVTs	NanoVelcro Microchip	EpCAM	Hoechst+/CK7+/HLA-G+/CD45−	3–6/2mL * 5–15/2 mL **	aCGH, STR
Huang CE (2017) [21]	24	11–13	EVTs	Cell Reveal with PicoBioChip	EpCAM	CK7+/ HLA-G+/CD45−/DAPI+	1–44/4mL	FISH, aCGH, STR, NGS
			fnRBCs	Cell Reveal^TM^ system with PicoBioChip	CD71	CD71+/GPA+/ CD45−/DAPI+	14–22/4mL	FISH, aCGH, STR, NGS
Feng (2018) [22]	13	~18	fnRBCs	Biotin-doped Ppy microchip	CD147	ε-HbF+/CD71+/DAPI+	NA	FISH, WEA
Ma (2019) [23]	14	13–27	fnRBCs	Cell Reveal^TM^ system with Coral Chip	CD71	CD71+/GPA+/CD45−/Hoechst+	2–71/2 mL	FISH, aCGH, STR, NGS
Sonek (2021) [24]	9	12–35	EVTs	LiquidScan^®^	EpCAM	None	1.28/mL	FISH
	6	12–35	EVTs	LiquidScan^®^	mixture	None	2.67/mL	FISH
Size-based (Label free)								
Cayrefourcq (2020) [25]	16	10–16	EVTs	Pasotix system	None	CD105+/ panCK+/ β-hCG+/CD45−	Only 5 fetal cells obtained	WGA
Huang Y (2020) [26]	30	NA	EVTs	CelutriateChip 1	None	CK7+/CD45−/DAPI+	1–3/2mL in 26/30 cases	WGA

HA/CTS NPs: biocompatible hydroxyapatite/chitosan nanoparticles; *: healthy cohort; **: disease cohort.

## Data Availability

Not applicable.

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
