# Peer review of "Recent Advances of Microfluidic Platform for Cell Based Non-Invasive Prenatal Diagnosis"

_ijms, 2023, doi:10.3390/ijms24020991_

Round 1

Reviewer 1 Report

• The primary output/endpoint variable(s)/measurement(s) of the study should be defined.  • Which randomization method was used in the distribution of the individuals included in the study to the groups? 

• Which blinding (masking) method was used in the study? 

• How was the sample size determined? This information should be explained in the Materials and Methods section. 

• Which sampling (probable or non-probable, etc.) method was used in the study?  • Statistical tests for hypothesis testing and their assumptions should be specified in the statistical analysis of the study in the Materials and Methods section.  • The details (version, license number, etc.) of the statistical package(s) or program(s) should be given in the section of "Data Analysis or Statistical Analysis".  • It should be explained how the qualitative and quantitative data are summarized under the sub-heading of Statistical Analyzes in the Materials and Methods section of the study.

Reviewer 2 Report

This article provides a short survey on the recent advances in microfluidic technology for noninvasive prenatal diagnosis (NIPD). The text discussed the progress of prenatal screening test based on the immunoaffinity microfluidic methods. The fetal cells to be isolated from  the peripheral blood of pregnant women include the fetal extravillous trophoblasts (fEVTs) and fetal nucleated red blood cells (fnRBCs).

The revised manuscript can consider the following comments to improve:

1) It is better to adjust the typesetting of Table 1 within one page.

2) In Section 4.1, it is suggesting expanding the presentation and discussions on microfluidic chip technology to detect and isolate circulating rare cells for NIPD. As it states in the text microfluidic chips are with "high purity, high sensitivity and high efficiency". It is necessary to describe the important experimental results of recent studies.

3) The mechanism of LCM should be provided in details, in order to interpret why such an approach can isolate the fetal cells with high purity.

4) Some minor mistakes, in Line 214, "Huang Y et al." should be "Huang et al." and Reference [27] is not cited in this sentence.

Round 2

Reviewer 1 Report

Acceptable

Reviewer 2 Report

The revised paper has been improved in response to the review comments. It is fine for publication.